# Bridge the Gap between SNN and ANN for Image Restoration

## Abstract

Models of dense prediction based on traditional Artificial Neural Networks (ANNs) require a lot of energy, especially for image restoration tasks. Currently, neural networks based on the SNN (Spiking Neural Network) framework are beginning to make their mark in the field of image restoration, especially as they typically use less than 10% of the energy of ANNs with the same architecture. However, training an SNN is much more expensive than training an ANN, due to the use of the heuristic gradient descent strategy. In other words, the process of SNN's potential membrane signal changing from sparse to dense is very slow, which affects the convergence of the whole model. To tackle this problem, we propose a novel distillation technique, called asymmetric framework (ANN-SNN) distillation, in which the teacher is an ANN and the student is an SNN. Specifically, we leverage the intermediate features (feature maps) learned by the ANN as hints to guide the training process of the SNN. This approach not only accelerates the convergence of the SNN but also improves its final performance, effectively bridging the gap between the efficiency of the SNN and the superior learning capabilities of ANN. Extensive experimental results show that our designed SNN-based image restoration model, which has only 1/300 the number of parameters of the teacher network and 1/50 the energy consumption of the teacher network, is as good as the teacher network in some denoising tasks.

## 1 Introduction

Image restoration, a classical research area in computer vision, focuses on recovering high-quality images from degraded observations. Most existing frameworks for image restoration use artificial neural networks (ANNs), which have high performance but also often rely on large-capacity models to achieve optimal performance. For instance, Restormer [47] and PromptIR [30] networks have 26.10M and 35.59M parameters, respectively, making them unsuitable for deployment on edge devices. The growing importance of devices with low power or battery constraints in various real-world applications, such as spiking neural networks (SNNs) offers a promising alternative [2, 6, 16, 36, 40, 41].

SNNs utilize binary signals (spikes) instead of continuous signals for neuron communication, reducing data transfer and storage overhead. Moreover, Spiking Neural Networks (SNNs) feature asynchronous processing and event-driven communication, which can eliminate redundant computations and synchronization burdens. When implemented in neuromorphic hardware, as mentioned in [26, 29], SNNs demonstrate exceptional energy efficiency. Unfortunately, in the challenging domain of image restoration, there is a notable absence of an SNN-based benchmark that can achieve performance levels comparable to those of its ANN counterpart. This is largely due to the slow training process of SNNs, which, relying on spike-based signaling, require extensive data exposure to generate predictions that match the accuracy of ANNs. This reliance on prolonged data exposure is particularly problematic when it comes to the extraction of subtle information from images that are visually redundant, as the training process becomes even more time-consuming.

In recent years, knowledge distillation as a promising approach for training heterogeneous models for knowledge transfer [11, 24, 28]. These efforts have piqued our curiosity, prompting us to explore the question: ***Is it feasible to transfer knowledge from ANNs to SNNs effectively?*** In this paper, our objective is to address the prolonged training times in SNNs by leveraging the exceptional perfor-

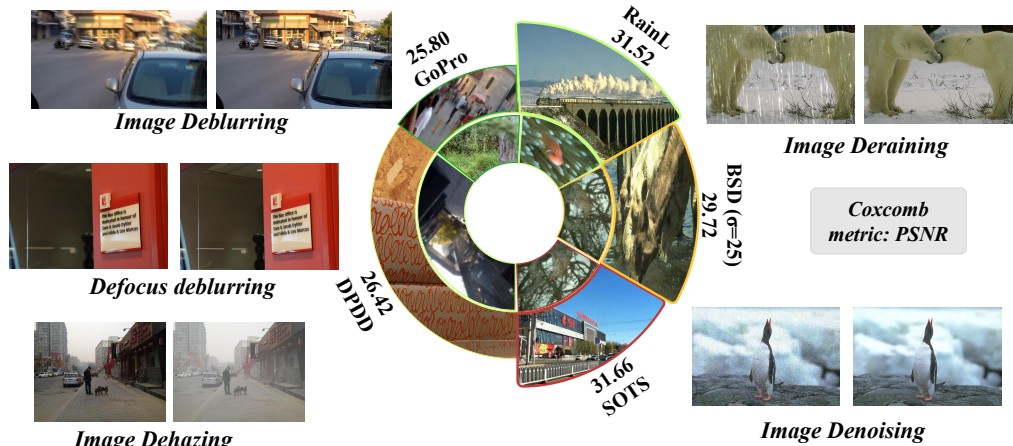

Figure 1: The Coxcomb of visual results and key evaluation metrics. Our SNN-based method transfers the knowledge from the ANNs for better image restoration performance

mance capabilities of ANNs to enhance and expedite the capabilities of SNNs. We propose a novel approach to train a thin SNN, called **H-KD**, which facilitates the distillation of knowledge in the feature space directly from ANNs. Specifically, we propose an efficient and effective SNN-based method, called **SpikerIR**, to solve the image restoration problem. This method aligns the representations from the ANN's decoder with those from our proposed SNN architecture, ensuring that the knowledge transferred is accurate information. Furthermore, considering that heterogeneous models may learn distinct predictive distributions due to their different inductive biases, we utilize the surrogate gradients to mitigate failure to surpass the performance of the original network (ANNs). Compared to other ANN-based deraining models, our method can gain better performance with shorter time steps. We consider the five different degradation types, as shown in Figure 1, helping produce visually appealing results across the different degradations.

Our main contributions are summarised as follows:

(1) We present SpikerIR, the first general image restoration SNN-based method to the best of our knowledge with a minimal parameter count of only **0.07**M, perfectly tailored for real-world applications on resource-limited devices.

(2) We design a scheme, called H-KD, to accelerate the SNNs training process, by distilling the knowledge from the ANN to obtain comparable performance for image restoration.

(3) Extensive experimental results on unified image restoration display that our proposed model can obtain excellent performance in a shorter time while reducing energy costs.

## 2 RELATED WORK

**Image Restoration.** Image restoration [42] focuses on reconstructing a degraded image to produce a high-quality version, addressing a core challenge in computer vision. This encompasses a range of tasks, including image denoising [50, 51], deraining [14, 33], dehazing [31, 34], and motion deblurring [5, 7] etc. Although these methods demonstrate remarkable reconstruction performance, their significant computational demands hinder their deployment in real-world applications, especially on resource-constrained devices. In contrast to these ANN-based methods, we introduce the use of SNN, which offers higher energy efficiency, as a framework for achieving effective and efficient image restoration.

**Deep Spiking Neural Networks.** Training strategies for deep SNNs primarily fall into two categories: direct training of SNNs and ANN-SNN conversion. Despite the promising advancements in directly training SNNs using techniques such as surrogate gradients and threshold-dependent batch normalization (TDBN) for deeper architectures, these approaches still suffer from several limitations. While methods like STBP [40] and subsequent works by [13] and Fang et al. [10] have

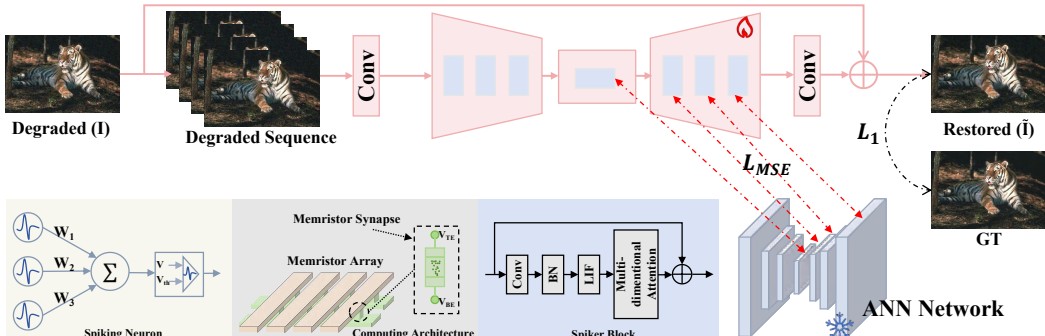

Figure 2: **Overview of our method.** SpikerIR transfers the knowledge from pre-trained ANNs' decoders to enhance the comprehension of degradation images, helping output high-quality content features. SpikerIR is designed as an encoder-decoder architecture, which mainly contains the Spiking Block with Spike Convolution Unit and Multi-dimensional Attention.

achieved success in classification tasks, they often require deep network structures (e.g., 50 layers) to perform well. This not only increases computational complexity but also contradicts the energy-efficient nature of SNNs. Furthermore, while there have been efforts to extend directly trained SNNs to regression tasks like object tracking [43, 49] and object detection [38], these approaches still rely on deep architectures to achieve satisfactory results. Song et al. [37] proposed efficient SNN architecture that has been implemented to remove rain from images. However, we tried to use it to achieve other image restoration tasks with unsatisfactory results. In contrast, our approach leverages artificial neural network feature distillation into SNNs, allowing SNNs to remain lightweight and power-efficient without sacrificing performance.

## 3 THE PRELIMINARIES OF SNNS

### 3.1 ENERGY CONSUMPTION

The number of operations is commonly used to estimate the computational energy consumption of neuromorphic hardware. In ANNs, each operation consists of multiplication and addition (MAC) involving floating-point numbers, and the computational burden is typically measured using floating-point operations (FLOPs)[1]. In contrast, SNNs offer an energy-efficient alternative for neuromorphic hardware, as neurons only engage in accumulation calculations (AC) when they spike. This efficiency allows SNNs to perform computations using a similar number of synaptic operations (SyOPs), significantly reducing energy consumption compared to traditional ANN architectures. We quantify the energy consumption of vanilla SNNs as $\mathbf{E}_{\text{SNN}} = \sum_n \mathbf{E}_b$, for each block $n$:

$$\mathbf{E}_b = \mathrm{T} \times (fr \times \mathbf{E}_{\text{AC}} \times \text{OP}_{\text{AC}} + \mathbf{E}_{\text{MAC}} \times \text{OP}_{\text{MAC}}), \tag{1}$$

where T and $fr$ represent the total time steps and the block firing rate. The blocks are normally convolutional or fully connected, and the energy consumption is determined by the number of AC and MAC operations ($\text{OP}_{\text{AC}}$, $\text{OP}_{\text{MAC}}$). In this work, we adopt the same structure of SNN and ANN to compare the energy consumption and assume that the data for various operations are 32-bit floating-point implementation in 45nm technology [12], where $\mathbf{E}_{\text{MAC}} = 4.6pJ$ and $\mathbf{E}_{\text{AC}} = 0.9pJ$.

### 3.2 STATIC IMAGE INPUTS

A common approach in SNNs for simulating pixel intensity signals in images is global encoding to generate spike signals. Taking into account the spatiotemporal properties of SNNs, we first apply direct encoding to the input degradation image to generate a sequence of spike trains, i.e. copying the single degraded image as the input for each time step $\mathbf{X} = \{\mathcal{X}_t\}_{t=1}^{\mathrm{T}}$ (in this paper, we set T to 4).

---

[1] https://github.com/sovrasov/flops-counter.pytorch

## 4 METHOD OVERVIEW

As illustrated in Figure 2, our method comprises two primary components: an encoder-decoder framework (student network) designed to learn features for capturing information, and the distillation of knowledge from the decoders of ANNs. These parts are collaboratively worked to improve the image restoration performance of SpikerIR.

Specifically, SpikerIR incorporates a lightweight encoder designed to extract degradation features from the degraded image sequence. To balance training efficiency and performance, we introduce 'prompts' derived from the ANN's network, where a prompt refers to the output of each ANN decoder layer, which guides SpikerIR's learning process. We employ Mean Squared Error (MSE) loss to align the outputs of each SNN decoder layer with those of the ANN, thereby facilitating learning from the ANN's output. Furthermore, to avoid the decreasing flexibility of our SpikerIR, we leverage the $L_1$ loss and an FFT-based frequency loss function, which is defined as:

$$L = \|\mathbf{I}_R - \mathbf{I}_{GT}\|_1 + \lambda \|\mathcal{F}(\mathbf{I}_R) - \mathcal{F}(\mathbf{I}_{GT})\|_1, \tag{2}$$

where $\mathbf{I}_R$ is the output of our model, $\mathbf{I}_{GT}$ is the high-quality ground-truth image, $\|\cdot\|_1$ denotes the $L_1$ loss, $\mathcal{F}$ represents the Fast Fourier transform, and $\lambda$ is a weight parameter that set to be 0.1 empirically. Algorithm 1 (H-KD training strategy) procedures can be written:

---

**Algorithm 1** the H-KD training strategy

---

**Input:**
    The output of each ANN's decoder layer $\mathbf{F}_{prompt}$;
    The output of each SpikerIR's decoder layer $\mathbf{F}_{optimize}$;
    The randomly initialized parameters $\mathbf{W}$ of a SpikerIR.
**Output:**
    The optimized parameters $\mathbf{W}^*$ of SpikerIR.

    $\mathbf{F}_{prompt} \leftarrow \{\mathbf{F}_{prompt}{}^{\mathbf{1}}, \ldots, \mathbf{F}_{prompt}{}^{\mathbf{i}}\}$;
    $\mathbf{F}_{optimize} \leftarrow \{\mathbf{F}_{optimize}{}^{\mathbf{1}}, \ldots, \mathbf{F}_{optimize}{}^{\mathbf{i}}\}$;
    $\mathbf{W}^* \leftarrow \gamma \underset{\mathbf{F}_{optimize}}{\arg\min} \mathcal{L}_{MSE}(\mathbf{F}_{optimize}, \mathbf{F}_{prompt}) + \underset{\mathbf{W}}{\arg\min} L_{Eq.(2)}(\mathbf{W})$;
    **return** $\mathbf{W}^*$;

---

where $\gamma$ represents the hyperparameter, which we set to 0.12 in this paper. It is worth noting that $\mathbf{F}_{prompt}$ and $\mathbf{F}_{optimize}$ may not match in the size of the feature maps, which can be aligned by interpolation and pooling.

## 5 EXPERIMENTS

We experimentally evaluate our method on five degradation types of tasks: *motion blurry*, *hazy*, *noise*, *rainy* and *defocus blurry*. In addition, we use three existing image restoration models as teacher networks to evaluate the effectiveness of our algorithm.

### 5.1 IMPLEMENTATION DETAILS

We train five sets of model parameters for these five image restoration tasks within the same network framework. Our SpikerIR uses a 4-layer encoder-decoder structure, where each layer of the network is a convolutional layer and a ReLU layer. From level-1 to level-4, the number of each level SpikerIR Blocks is 2, and the number of channels is $\{48,96,192,384\}$. For teather networks, we adopt Restormer [46], PromptIR [30] and AdaIR [8] as the teacher models. We train models with AdamW optimizer ($\beta_1$=0.9, $\beta_2$=0.999, weight decay 0.05) for 51 epochs with the initial learning rate 0.0005 gradually reduced to 0.00001 with the cosine annealing for image denoising tasks. Distinct tasks, however, were trained for different numbers of epochs to optimize performance, with the details as follows: Motion deblurring for 77 epochs, dehazing for 5 epochs, deraining for 8 epochs, and defocus deblurring for 208 epochs. We start training with patch size $64 \times 64$ and batch size 8. For data augmentation, we use horizontal and vertical flips. Two well-known metrics, Peak Signal-to-Noise Ratio (PSNR) and Structural Similarity (SSIM), are employed for quantitative comparisons. Higher values of these metrics indicate superior performance of the methods.

## 5.2 MAIN RESULTS

We show some quantitative and qualitative results. It is worth noting that some teacher models were not trained on the specific dataset, so the results shown may only have one SpikerIR. Some reports have three teacher models, and the results shown will have three SpikerIR's, which are represented as three student networks of teacher networks.

**Image Denoising Results.** We conduct denoising experiments on the synthetic benchmark dataset BSD68 [25], generated using additive white Gaussian noise. Table 1 presents the result for color image denoising. In alignment with previous methods [23, 52], we evaluated noise levels of 15, 25, and 50 during testing. Our SpikerIR achieves excellent performance for denoising tasks. Additionally, for the noise level 15 and 25, SpikerIR surpasses the teacher model Restormer. Figure 3 shows the denoised results by feature model and Our SpikerIR correspondingly for color denoising.

Table 1: **Single-image motion denoising** results. The H-KD method is applied to three different methods, i.e. Restormer, PromptIR, and AdaIR.

| Method | $\sigma=15$ | | $\sigma=25$ | | $\sigma=50$ | |
|---|---|---|---|---|---|---|
| | PSNR | SSIM | PSNR | SSIM | PSNR | SSIM |
| Restormer | 31.96 | 0.900 | 29.52 | 0.884 | 26.62 | 0.688 |
| SpikerIR | 32.35 | 0.894 | 29.72 | 0.825 | 26.13 | 0.678 |
| PromptIR | 33.98 | 0.933 | 31.31 | 0.888 | 28.06 | 0.799 |
| SpikerIR | 32.48 | 0.898 | 29.70 | 0.829 | 25.59 | 0.642 |
| AdaIR | 34.12 | 0.935 | 31.45 | 0.892 | 28.19 | 0.802 |
| SpikerIR | 32.29 | 0.889 | 29.53 | 0.817 | 25.34 | 0.623 |

**Image Deraining Results.** The PSNR and SSIM scores across the RGB channels, as shown in Table 2, demonstrate that while our SpikerIR model achieves lower scores compared to state-of-the-art methods such as Restormer, PromptIR, and AdaIR, it is important to note that SpikerIR operates with only 1/300 of the parameter count. Our SpikerIR model adopts a similar architecture to Restormer, making Restormer a natural choice as the teacher model for comparison. Consequently, Restormer outperforms both PromptIR and AdaIR in this context, as its architectural design aligns more closely with that of SpikerIR. This alignment enables Restormer to serve as a more effective reference for evaluating SpikerIR's performance.

**Single-image Motion Deblurring Results.** Here, we use only Restormer as the teacher network, due to PromptIR and AdaIR were not evaluated poorly on this dataset. We evaluate deblurring methods both on the synthetic dataset (GoPro [27]) and the real-world datasets (RealBlur-R [35], RealBlur-J [35]). Table 3 shows that our SpikerIR receives a similar performance as the SOTA ANN models with fewer parameters and lower complexity.

Table 2: **Image deraining** results. The H-KD method is applied to three different methods, i.e. Restormer, PromptIR, and AdaIR.

| Method | Test100 [48] | | Rain100H [44] | | Rain100L [44] | |
|---|---|---|---|---|---|---|
| | PSNR | SSIM | PSNR | SSIM | PSNR | SSIM |
| Restormer [46] | 32.00 | 0.923 | 31.46 | 0.904 | 38.99 | 0.978 |
| SpikerIR | 28.20 | 0.854 | 29.06 | 0.810 | 34.90 | 0.938 |
| PromptIR | 30.23 | 0.901 | 30.88 | 0.877 | 37.44 | 0.979 |
| SpikerIR | 30.22 | 0.902 | 30.15 | 0.856 | 33.71 | 0.934 |
| AdaIR | 31.79 | 0.979 | 30.99 | 0.889 | 38.02 | 0.981 |
| SpikerIR | 31.55 | 0.973 | 28.64 | 0.799 | 34.51 | 0.924 |

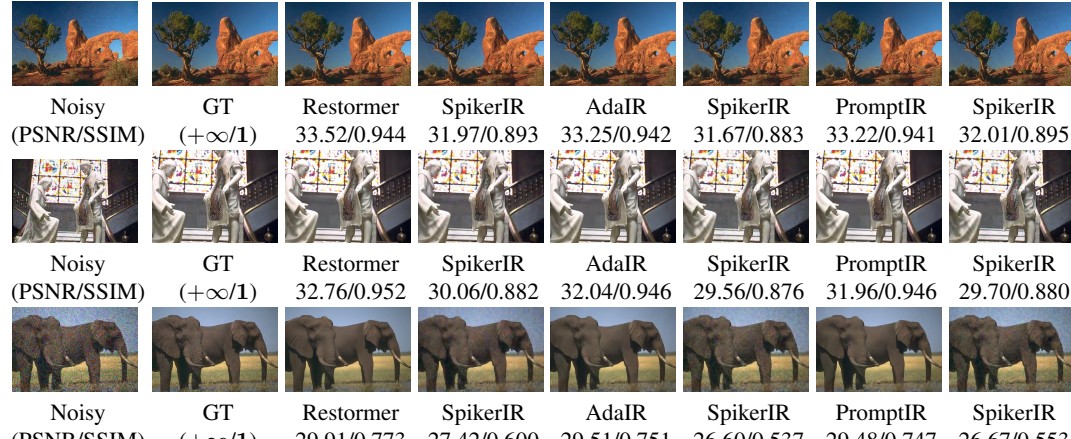

Figure 3: Visual results on **Image denoising**. Top row: the noise level is 15. Middle row: the noise level is 25. Bottom row: the noise level is 50.

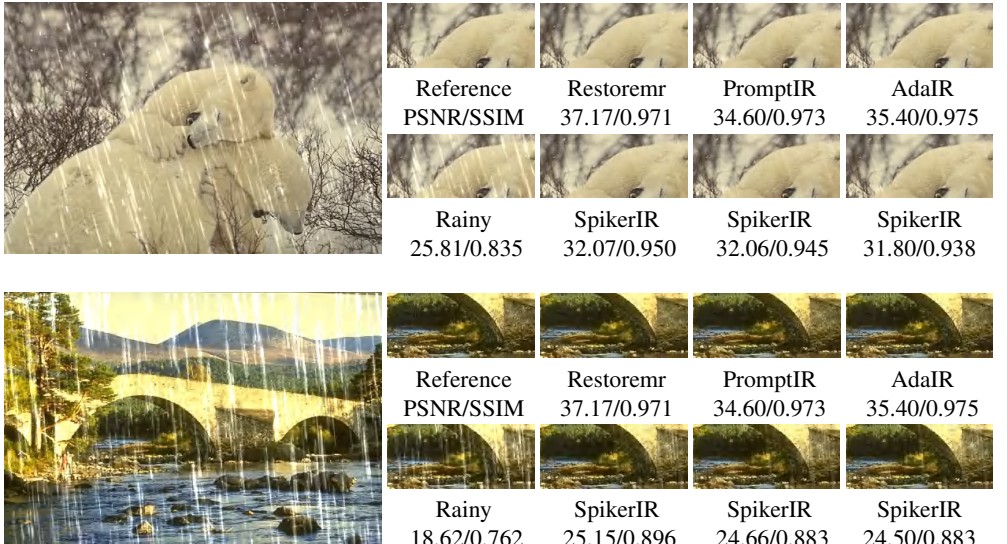

Figure 4: Single image deraining. Compared to the teacher model, our SpikerIR achieves comparable performance with significantly fewer parameters.

Table 3: Single image motion deblurring results. Our method was compared with other motion deblurring methods, and it achieved superior results on the RealBlur dataset. In addition, the number of parameters and FLOPS are much smaller than those of other models.

| Method | GoPro [27] PSNR SSIM | RealBlur [35] PSNR SSIM | Params (M) | FLOPs (G)) |
|---|---|---|---|---|
| MPRNet [45] | 32.66 0.959 | 29.65 0.892 | 20.10 | 777.01 |
| MIMO-UNet++ [4] | 32.68 0.959 | 33.37 0.856 | 617.64 | 16.10 |
| Restormer [46] | 32.92 0.961 | 33.69 0.863 | 26.10 | 12.33 |
| Stripformer [39] | 33.08 0.962 | 25.97 0.866 | 20.0 | 170.46 |
| SpikerIR (Ours) | 29.89 0.931 | 30.25 0.899 | **0.07** | **0.03** |

**Image Defocus Deblurring Results.** Table 4 reports the performance of our SpikerIR on the image defocus deblurring task. Figure. 6 shows that our SpikerIR has a comparable performance in terms of deblurring quality. We also present zoomed-in cropped patches in yellow and green boxes.

**Image Dehazing Results.** We evaluate SpikerIR on the synthetic dataset (RESIDE/SOTS) [20]. Compared to PromptIR [30], our method generates a 0.35 dB PSNR improvement. As the Figure 7

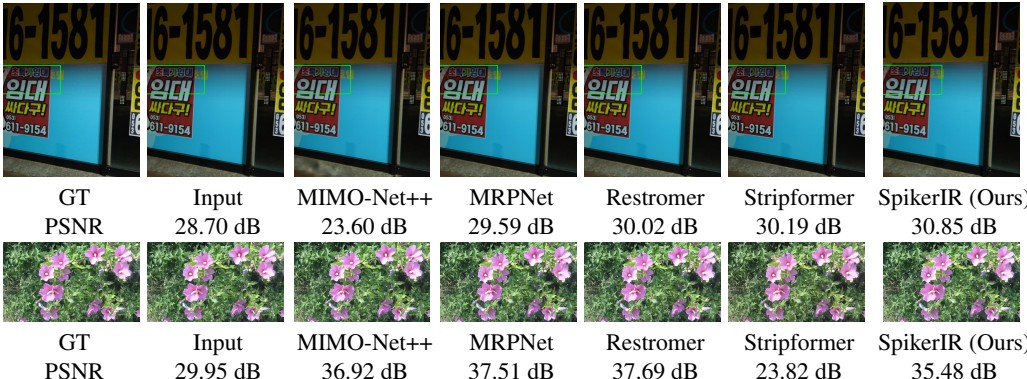

| | | | | | | |
|---|---|---|---|---|---|---|
| GT | Input | MIMO-Net++ | MRPNet | Restromer | Stripformer | SpikerIR (Ours) |
| PSNR | 28.70 dB | 23.60 dB | 29.59 dB | 30.02 dB | 30.19 dB | 30.85 dB |

| | | | | | | |
|---|---|---|---|---|---|---|
| GT | Input | MIMO-Net++ | MRPNet | Restromer | Stripformer | SpikerIR (Ours) |
| PSNR | 29.95 dB | 36.92 dB | 37.51 dB | 37.69 dB | 23.82 dB | 35.48 dB |

Figure 5: Visual results on image deblurring. Top row: Realworld deblurring on RealBlur dataset. Bottom row: Synthetic deblurring on Gopro Dataset.

Table 4: **Defocus deblurring** comparisons on the DPDD testset [1] (containing 37 indoor and 39 outdoor scenes). Our SpikerIR achieves excellent performance.

| | Indoor Scenes | | | | Outdoor Scenes | | | | Combined | | | |
|---|---|---|---|---|---|---|---|---|---|---|---|---|
| Method | PSNR | SSIM | MAE | LPIPS | PSNR | SSIM | MAE | LPIPS | PSNR | SSIM | MAE | LPIPS |
| EBDB [15] | 25.77 | 0.772 | 0.040 | 0.297 | 21.25 | 0.599 | 0.058 | 0.373 | 23.45 | 0.683 | 0.049 | 0.336 |
| DMENet [17] | 25.50 | 0.788 | 0.038 | 0.298 | 21.43 | 0.644 | 0.063 | 0.397 | 23.41 | 0.714 | 0.051 | 0.349 |
| DPDNet [1] | 26.54 | 0.816 | 0.031 | 0.239 | 22.25 | 0.682 | 0.056 | 0.313 | 24.34 | 0.747 | 0.044 | 0.277 |
| IFAN [18] | 28.11 | 0.861 | 0.026 | 0.179 | 22.76 | 0.720 | 0.052 | 0.254 | 25.37 | 0.789 | 0.039 | 0.217 |
| Restormer [46] | 28.87 | 0.882 | 0.025 | 0.145 | 23.24 | 0.743 | 0.050 | 0.209 | 25.98 | 0.811 | 0.038 | 0.178 |
| SpikerIR (Ours) | 26.42 | 0.801 | 0.030 | 0.287 | 21.50 | 0.648 | 0.059 | 0.411 | 23.89 | 0.727 | 0.045 | 0.351 |

Table 5: Dehazing results in the single-task setting on the SOTS-Outdoor [20] dataset.

| Method | DehazeNet [3] | MSCNN [34] | AODNet [19] | EPDN [32] | FDGAN [9] | AirNet [21] | Restormer [46] | PromptIR [30] | AdaIR [8] | SpikerIR (Ours) |
|---|---|---|---|---|---|---|---|---|---|---|
| PSNR | 22.46 | 22.06 | 20.29 | 22.57 | 23.15 | 23.18 | 30.87 | 31.31 | 31.80 | 31.66 |
| SSIM | 0.851 | 0.908 | 0.877 | 0.863 | 0.921 | 0.900 | 0.969 | 0.973 | 0.981 | 0.975 |

shown, our SpikerIR is effective in removing degradations and generates images that are visually closer to the ground truth.

# 6 ABLATION STUDY AND APPLICATION

In this section, we train Gaussian color denoising models on image patches of size $64 \times 64$ for 51 epochs only. Testing is performed on BSD68 [25] dataset. Flops and energy statistics are computed on image size $256 \times 256$. The feature models that we selected are the well-known Restormer [46], PromptIR [30], and AdaIR [8].

**Impact of knowledge distillation** In our H-KD method, the ANN teacher model's decoder features are integrated into the SpikerIR to enhance intermediate-level learning. We conduct experiments to understand the impact of aligning intermediate layer features from the ANN teacher model on our SpikerIR's performance during knowledge distillation. As the SpikerIR is the encoder-decoder framework, we perform three ablation experiments to evaluate the effects of feature constraints at different stages (stages: $1 \rightarrow 7$). First, we apply constraints to all the encoder and decoder layers. Second, we restrict the constraints to stages $3 \rightarrow 5$. Finally, we only constrain the decoder, i.e. stages $4 \rightarrow 7$. As shown in Table 6, when comparing learning at different feature ANNs, the SpikerIR student presents no preference for different features on denoising tasks, as they all help improve the performance.

**Performance Comparison with Equivalent ANNs** To highlight the efficiency of our SNN model, we compare its performance with equivalent ANNs, trained and tested using the same strategies on the BSD dataset. This comparison allows us to evaluate the energy consumption differences between

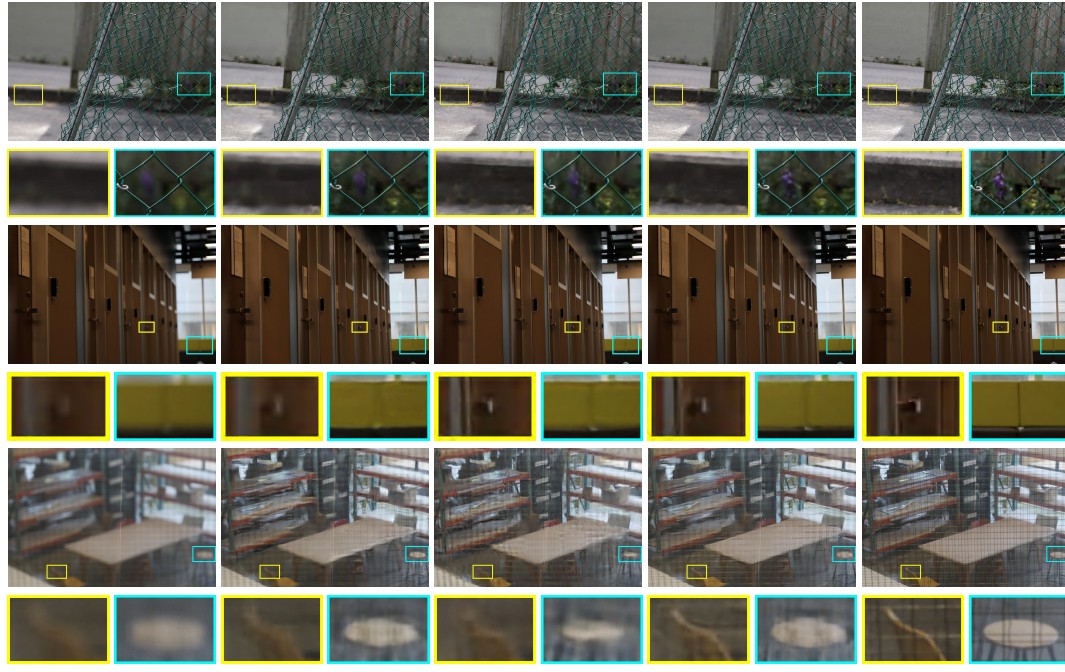

(a) Input  (b) IFAN () [18] (c) Restormer [46] (d) SpikerIR (Ours) (e) GT

Figure 6: Qualitative comparisons with IFAN [18] and Restormer [46] on the test set of the DPDD dataset [1] for image defocus deblurring.

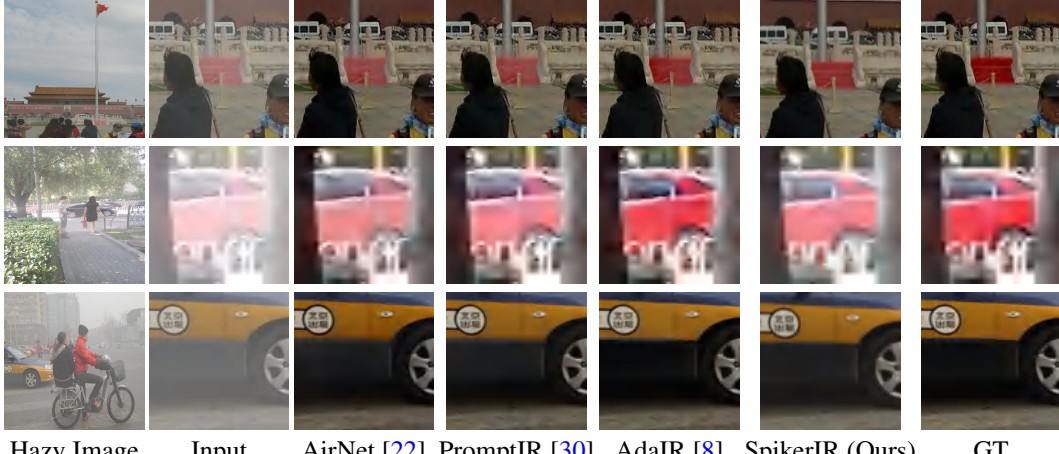

Hazy Image  Input  AirNet [22] PromptIR [30] AdaIR [8] SpikerIR (Ours)  GT

Figure 7: Image dehazing comparisons under the single task setting on SOTS [20].

the models while maintaining comparable performance. Specifically, for the ANN model, we set the number of encoder and decoder layers to match those of our SpikerIR model, and we adopt the Restormer framework for the architecture. Table 7 reports the performance, parameters and energy consumption comparison between our SNN model and the ANN model for the equivalent architecture. The evaluation models were run on a Lynxi HP300 platform to demonstrate the performance of the models. Our model is trained on dehazing and deraining datasets with the accuracy of float16. As the Figures 9 and 8 shown, our method has better visualization, especially on real-world image dehazing tasks.

## 7 DISCUSSION AND LIMITATIONS

By leveraging the intermediate features from the ANN teacher model, we successfully accelerate the convergence of the SNN student (the aggregation speed of SNN models is increased by more than

Table 6: Ablation experiments on image denoising for learning the impact of aligning intermediate features. Without the use of distillation, there is a significant reduction in the performance of the student network.

| Teacher Model | Stage $1 \rightarrow 7$ | | | Stage $3 \rightarrow 5$ | | | Stage $3 \rightarrow 5$ | | | w/o KD | | |
|---|---|---|---|---|---|---|---|---|---|---|---|---|
| | $\sigma=15$ | $\sigma=25$ | $\sigma=50$ | $\sigma=15$ | $\sigma=25$ | $\sigma=50$ | $\sigma=15$ | $\sigma=25$ | $\sigma=50$ | $\sigma=15$ | $\sigma=25$ | $\sigma=50$ |
| Restormer | 32.45 | 29.65 | 25.68 | 31.36 | 29.00 | 25.07 | 32.35 | 29.72 | 26.13 | 30.46 | 27.91 | 22.35 |
| PromptIR | 32.50 | 29.61 | 25.04 | 31.98 | 29.15 | 25.35 | 32.48 | 29.70 | 25.59 | 31.44 | 25.00 | 23.26 |
| AdaIR | 32.33 | 29.48 | 25.22 | 31.99 | 29.33 | 25.16 | 32.29 | 29.53 | 25.34 | 31.29 | 25.11 | 22.89 |

Table 7: Comparison with Equivalent ANNs on image denoising.

| Teature Model | Student Model | BSD68 | | | Flops G | Params M | Energy uJ |
|---|---|---|---|---|---|---|---|
| | | $\sigma=15$ | $\sigma=25$ | $\sigma=50$ | | | |
| Restormer | SpikerIR | 32.45 | 29.65 | 25.68 | 0.07 | 0.03 | $5.232 \times 10^4$ |
| | ANN | 33.00 | 30.33 | 26.87 | 10.22 | 71.18 | $7.331 \times 10^6$ |

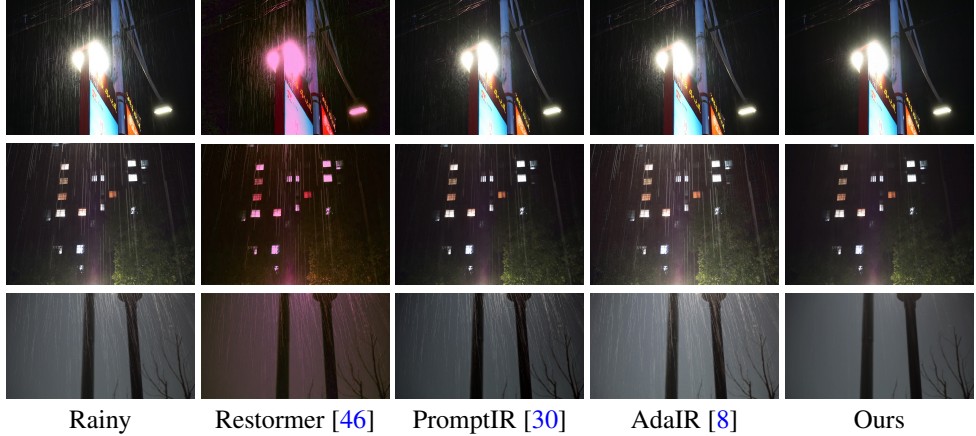

| Hazy | Restormer [46] | PromptIR [30] | AdaIR [8] | Ours |
|---|---|---|---|---|

Figure 8: Real-world image dehazing results. Our method recovers images with more contrast and the haze is effectively removed.

| Rainy | Restormer [46] | PromptIR [30] | AdaIR [8] | Ours |
|---|---|---|---|---|

Figure 9: Real-world image deraining results. Our method effectively removes rain streaks and the glow of the wick is effectively limited.

$5 \times$), reducing the computational burden typically associated with training SNNs. However, during our experiments, we observed two limitations related to inference and training time.

i) The inference time in the SNN model is slower compared to its ANN counterpart (on platforms without SNN-optimized hardware, see Figure 10(b)). Although SNNs offer significant energy savings, their event-driven nature and reliance on temporal dynamics during the inference process introduce latency. Future work could explore more optimized spiking neuron models or hybrid approaches that combine the advantages of both SNNs and ANNs to improve inference speed without sacrificing energy efficiency.

ii) Training time per epoch increases progressively as training continues, particularly in later stagessee Figure 10(a). This phenomenon is primarily due to the complexity of gradient-based optimization in SNNs, where updating spiking neurons' membrane potentials becomes more computationally demanding as the model learns. Addressing this issue will require the development of more efficient training algorithms or hardware-accelerated solutions specifically designed for SNNs.

In addition, Figures 10(c) and 10(d) show the voltage shift of the SNN with the H-KD strategy, where the shift in Figures 10(c) is more drastic and non-sparse. A denser voltage can more effectively extract the features of an image.

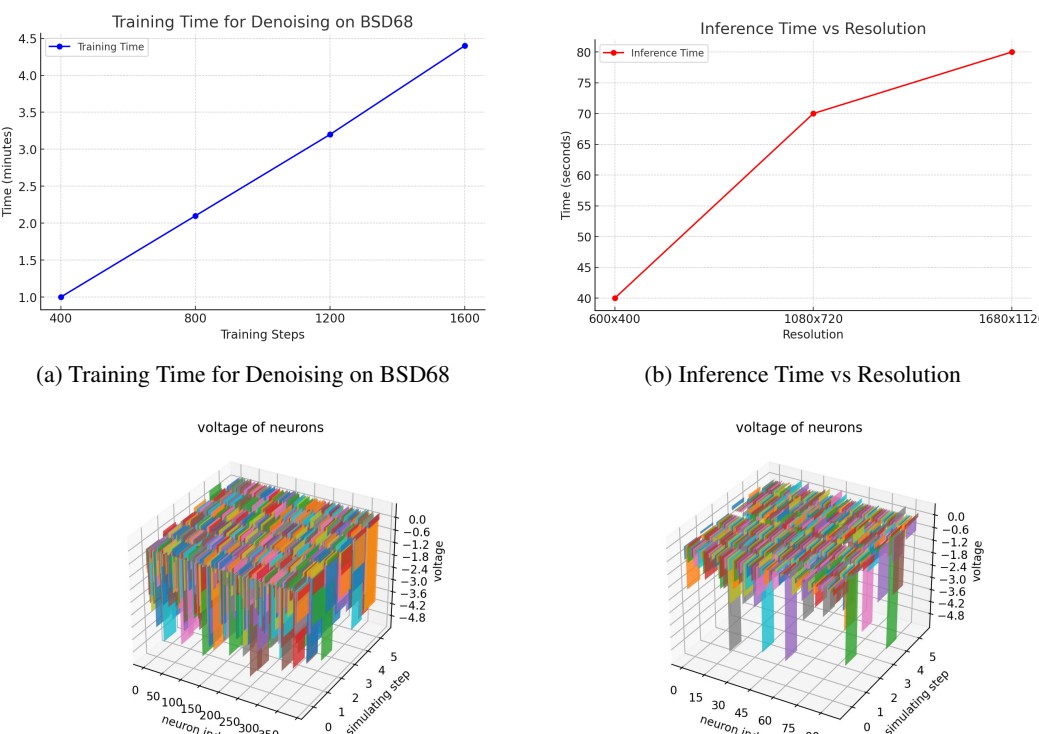

(a) Training Time for Denoising on BSD68

(b) Inference Time vs Resolution

(c) The voltage visualization of SpikerIR with the H-KD method.

(d) The voltage visualization of SpikerIR without the H-KD method.

Figure 10: Comparison of training and inference times and the state of voltage membrane changes in the SNN during training.

## 8 CONCLUSION

In this paper, we develop an efficient, low-energy network named SpikerIR for a variety of image restoration tasks. On both GPU platforms and embedded platforms, our model demonstrates excellent performance, with clearer images than those recovered by the teacher network on some datasets. In addition, we attempted to explain the role of distillation, which acts to enable the conversion of a sparse voltage to a denser one. We also discuss some of the limitations of the model, in particular the fact that SNNs rely heavily on efficient I/O operations.

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
