# OpenReview forum: "Bridge the Gap between SNN and ANN  for Image Restoration"
_ICLR.cc/2025/Conference — ICLR 2025 Conference Withdrawn Submission_

### Official Review · Reviewer_Bwhg · 2024-10-30

**Soundness:** 3
**Presentation:** 3
**Contribution:** 2
**Rating:** 3
**Confidence:** 5

**Summary:**

The paper focuses on the application of Spiking Neural Networks (SNNs) in the field of image restoration. It highlights the limitations of traditional Artificial Neural Networks (ANNs), which, while effective, often require large model capacities that are unsuitable for deployment on low-power devices. SNNs present a promising alternative due to their event-driven nature and energy efficiency. The authors propose a Heterogeneous Knowledge Distillation to address the prolonged training times associated with SNNs. This method facilitates the transfer of knowledge from ANNs to SNNs, allowing the latter to achieve better performance in image restoration tasks. The proposed SNN-based method, named SpikerIR, aligns the representations from the ANN's decoder with those from the SNN architecture, ensuring accurate knowledge transfer. The paper presents both quantitative and qualitative results demonstrating the effectiveness of SpikerIR in various image restoration tasks.

**Strengths:**

1.	SNNs are inherently more energy-efficient than ANNs, making them suitable for deployment on edge devices with power constraints.
2.	The paper provides both quantitative and qualitative results, showcasing the performance of the proposed method across different image restoration tasks, which strengthens the validity of the findings.

**Weaknesses:**

1.	Actually, as reported in Table 1 and Table 2, compared to SOTA ANN-based models, Restormer, PromptIR, and AdaIR, the PSNR/SSIM performance of your SpikerIR is limited, such as 31.55 dB vs. 31.79 dB in Test100 dataset. I wonder if you can upgrade the performance of SNN-based network by adding some CNN-based attention mechanism, such as channel attention or spatial attention.
2.	Missing in-depth motivation. This article seems to be just a simple attempt at an IR mission by SNN. As demonstrated in the experimental section, SNN-based model can not achieve comparative performance against CNN or Transformer-based model, which raise a critical concern if SNN is suitable for IR task? The author should add more in-depth motivation of using SNN by referring some published works or some experimental finds.
3.	The effectiveness of the proposed SNN model is heavily reliant on the quality of the ANN from which it is distilling knowledge. If the ANN performs poorly, the SNN may also struggle to achieve satisfactory results. I suggest the author conduct more experiments demonstrating how their method performs with ANNs of varying quality as teachers, or propose a discussion on strategies to mitigate this dependency.
4.	Missing real-world evaluation or deployment. For instance, NIQE results and visual comparison should be added. Besides, I wonder if you approach can be deployed to some real-world devices to demonstrate the efficiency of SNN against traditional IR model.

**Questions:**

See the weakness.

---

### Official Review · Reviewer_byRX · 2024-11-02

**Soundness:** 2
**Presentation:** 3
**Contribution:** 3
**Rating:** 5
**Confidence:** 4

**Summary:**

This paper introduces SpikerIR, an SNN-based image restoration model that leverages knowledge distillation from ANNs to enhance performance with fewer parameters and lower energy consumption. The proposed H-KD training strategy accelerates SNN convergence and improves final output, bridging the efficiency gap between SNNs and ANNs. Despite slower inference times, SpikerIR demonstrates competitive results in denoising tasks, making it a promising solution for energy-efficient image restoration on edge devices.

**Strengths:**

1. The SpikerIR model achieves comparable performance to ANNs with significantly reduced parameters and energy consumption, demonstrating higher energy efficiency in image restoration tasks compared to traditional ANNs.

2. The proposed H-KD training strategy enables rapid learning from ANNs and speeds up the training process of SNNs, enhancing model convergence, which is a significant advancement in the field since training SNNs is typically more time-consuming and costly than training ANNs.

**Weaknesses:**

1. Although this paper has achieved some level of energy efficiency improvement and faster model convergence for SNNs in image restoration, the overall degree of innovation in the method is not high, as its main ideas have already been validated in other computer vision tasks.
2. How does this paper ensure a fair comparison with comparison methods? For instance, as far as I know, the choice of patch size can affect experimental results to some extent. Has an ablation study been conducted?
3、In this paper, the phrase "were trained for different numbers of epochs to optimize performance," was this determined adaptively or set manually?
4. The paper lacks comparisons with the latest methods, such as experimental comparisons with the method from reference [37].
5. The paper has some grammatical and formatting errors or inconsistencies, such as: in Table 3, the unit for Flops should be (G) instead of (G)), and there is a lack of uniformity in the formatting of conference references.
6. The author needs to further explain why comparisons were made with other specific models in other tasks, such as deblurring, but not in the rain removal task.
7. In addition, as is well known, there is a significant problem with using the ANN-SNN strategy, which is that the performance of SNN heavily depends on the performance of ANN and cannot surpass ANN. I am curious about how the author solved this problem. The experimental section seems to lack discussion in this regard.

**Questions:**

Please see the Weaknesses

---

### Official Review · Reviewer_ANnG · 2024-11-04

**Soundness:** 2
**Presentation:** 2
**Contribution:** 1
**Rating:** 3
**Confidence:** 4

**Summary:**

This work focuses on designing efficient SNN-based models for image restoration with knowledge distillation. To this end, the authors proposed asymmetric framework distillation by transferring knowledge from the teacher ANN to the student SNN. Experiments show that the proposed model has only 1/300 the number of parameters of the teacher network.

**Strengths:**

1. The method is simple yet effective.
2. The proposed methods has much less parameters and energy consumption.

**Weaknesses:**

1. The novelty is very limited by using knowledge distillation, which has been widely explored in various tasks.
2. Experiments are not convincing enough. For example in Table 1, the student SpikerIR, surprisingly, has higher PSNR values than the teacher Restormer on \sigma=15.

**Questions:**

1. What are the advantages of SNN-based IR methods?
1. What are the effects of other knowledge distillation methods?

---

### Official Review · Reviewer_5KQj · 2024-11-04

**Soundness:** 2
**Presentation:** 3
**Contribution:** 3
**Rating:** 5
**Confidence:** 3

**Summary:**

This paper propose a novel distillation technique, called asymmetric framework (ANN-SNN) distillation, in which the teacher is an ANN and the student is an SNN.

**Strengths:**

The author introduce SNN for restoration, which is interesting.

**Weaknesses:**

1.The performance of SNN is significantly worse than that of Restormer, as shown in Tables 2, 3, and 4. I hope the authors can increase the parameter count to see how much is needed to bring SNN's performance in line with Restormer. I suspect that SNN's upper limit will not exceed that of Restormer.

2.I hope the authors will not test FLOPs; please test the actual running times of Restormer and SNN, and report the devices and other settings used. It is well known that FLOPs are not meaningful for comparing different computation methods.

3.I hope the authors will compare SNN with other SOTA lightweight restoration methods, such as distillation, pruning, or model lightweight sparsity methods.

4. It seems that there is no officially open-source dehaze model for Restormer; did you train it yourself?

**Questions:**

see above

---

### Note · Authors · 2024-12-18

I have read and agree with the venue's withdrawal policy on behalf of myself and my co-authors.